# Corrosion Activity of Carbon Steel B450C and Low Chromium Ferritic Stainless Steel 430 in Cement Extract Solution

**Ángel Bacelis** [1] , **Lucien Veleva** [1,*] , **Sebastián Feliu, Jr.** [2] , **Marina Cabrini** [3] and **Sergio Lorenzi** [3]

1   Applied Physics Department, Center for Investigation and Advanced Study (CINVESTAV),
    Merida 97310, Mexico; angel.bacelis@cinvestav.mx
2   Nacional Center for Metallurgical Research (CENIM-CSIC), Surface Engineering,
    Corrosion and Durability Department, 28040 Madrid, Spain; sfeliu@cenim.csic.es
3   Department of Engineering and Applied Sciences, INSTM RU Bergamo and University of Bergamo,
    Viale Marconi 5, 24044 Dalmine, Italy; marina.cabrini@unibg.it (M.C.); sergio.lorenzi@unibg.it (S.L.)
*   Correspondence: veleva@cinvestav.mx; Tel.: +52-999-942-9400

**Abstract:** This study compares corrosion activities of carbon steel B450C and SS 430 (Mn in low content) exposed for 30 days in cement extract solution. Iron oxide and hydroxide were formed as corrosion products, in addition to $CaCO_3$, in the presence of $Cr_2O_3$ on SS 430. Because of the decrease in pH, B450C lost the passive state when OCP shifted to negative values, while SS 430 showed positive OCP values, maintaining its passive state. The SEM images confirmed that the corrosion attack on the surface was less aggressive for SS 430. The Nyquist plots of EIS initially showed capacitive behavior and later changed to semi-linear diffusion impedance, which SS 430 maintained firmly. The phase angle Bode diagrams confirmed these changes. Two equivalent circuits were applied. The calculated values of $R_p$ for SS 430 increased over time (protective passive layer mainly of $Cr_2O_3$ oxide), while for carbon steel, $R_p$ reached maximum value after 168 h and then decreased, maintaining minimum values approximately five orders lower than those of the stainless steel.

**Keywords:** carbon steel; stainless steel; concrete pore; cement extract; corrosion tests; XPS

## 1. Introduction

Concrete is one of the most widely used materials in the construction industry because of low cost and mechanical performance [1]. Concrete supports compressive stresses; however, it is susceptible to cracking due to other types of mechanical stresses such as bending, traction, torsion, and shear, among others. Steel bars to reinforce concrete provides resistance to these forces because of the steel-concrete union [2]. To confront the highly aggressive environments, stainless steels (Fe-Cr based alloys) have been proposed as reinforcement on account of their excellent mechanical properties and high resistance to corrosion, which guarantee the durability and service life of the reinforced concrete structure [3].

The stability of carbon steel reinforcement lies in the formation of passivation layers of oxides/oxyhydroxides of nanometric thickness (circa 20 nm), responsible for the very low corrosion rate of the reinforcing bars in their passive state [4,5]. Usually, this layer is composed of an inner one rich in Fe (II) oxides/oxyhydroxides and an outer one containing Fe (III) oxides/oxyhydroxides [6,7]. The high corrosion resistance of stainless steels (SS) results from the formation of a continuous, protective surface oxide layer, giving rise to the passive state of the metal surface [8,9]. The native oxide formed on Fe-Cr-based stainless steel surfaces generally shows a duplex structure, with an iron-rich outer layer and a chromium-rich inner layer [6,7,9–12]. The chromium-rich oxide plays a key role in the corrosion resistance [13–15].

The passive steel state disappears when the passivation layer loses thickness because of the initiation of the corrosion process, which leads to a loss of steel-concrete adhesion and,

therefore, to a loss in the mechanical properties of reinforced concrete; this is accompanied by micro- and macro-cracks as well as localized pitting penetration in the SS passive layer. Thus, the thickening of the corrosion layer may promote the detachment of the concrete cover because of the mechanical internal stresses of corrosion products of iron formed on the carbon steel surface, which are usually voluminous [16].

According to the Pourbaix diagram, the passivity of the carbon steel occurs naturally in alkaline media [17]. Portland cement, Portlandite ($Ca(OH)_2$), provides the necessary highly alkaline environment in concrete [4].

The concepts of pH and steel passivity require the presence of alkaline concrete pore solution, as a hydration product of cement (pH > 12.5), when the structure is exposed to natural environments, considering that the concrete pores (from micrometer to nanometer size [18,19]) could maintain that same pH of aqueous solution.

It is well known that cathodic reaction plays an important role in the establishment of the equilibrium of a corrosion process. In alkaline and neutral electrolytes, the reaction occurs as oxygen reduction (Equation (1)) over time at the steel-concrete surface, and it is suggested that this reaction mainly occurs on oxidized porous surfaces, which are complex in composition and in morphology [20]. In this respect, the decrease in the amount of the diffused $O_2$ and an increase in $OH^-$ ion content may inhibit the reaction.

$$O_2 + 2H_2O + 4e^- \longleftrightarrow 4OH^- \tag{1}$$

Furthermore, as the ferrous iron is soluble in water at any pH [21], the hydrolysis of free ferrous ions may cause a local diminishing of pH as well as oxide reduction, responsible for the passive iron state. On the other hand, the alkaline pH value may diminish because of $CO_2$ ambient pollution when a phenomenon known as carbonation occurs, as a transformation of $Ca(OH)_2$ into $CaCO_3$ [4,18,22].

As an important part of the internal environment of concrete, the pore solution plays a considerable role in preventing or accelerating the steel corrosion [23]. A variety of alkaline solutions simulating the electrolytic environments of concrete pores have been used in order to evaluate the electrochemical behavior of steel-reinforced concrete over a short period of time, although the proposed models differ in the results obtained [24–26]. Furthermore, a major concern is the composition of alkaline solutions, which may influence the corrosion resistance of steels, as well as the composition of the corrosion products formed [26–32]. Few works have proposed the use of cement extract solution, which may provide variety with regards to the ions found in the concrete pores in real conditions [33–37]. There is still considerable controversy about the effect of the composition of the model solution on the electrochemical behavior of steels [38].

This research compares the corrosion activity of commercial Italian carbon steel to that of low chromium ferritic Finnish stainless steel, exposed for 30 days in cement extract unbuffered solution in order to simulate the concrete environment at the steel-concrete-pore interface. Both steels have been proposed as reinforcement in a lower pH of concrete than the traditional pH of Portland cement-concrete, in the presence of binders. However, it is very important to first establish the corrosion activity of each steel when pH in the traditional concrete-pore environment changes in time. Applying a variety of different techniques and methods help to contribute in this aspect. Two non-destructive electrochemical techniques were performed: free corrosion potential monitoring at open circuit potential (OCP) and electrochemical impedance spectroscopy (EIS). The surfaces of the steels were characterized by scanning electron microscopy (SEM) and X-ray photoelectron spectroscopy (XPS) techniques. To our knowledge, no other research on this topic has been previously undertaken.

## 2. Materials and Methods

### 2.1. Samples and Solution Preparation

Flat samples of carbon steel construction material (B450C), supplied by Pittini Group (Gemona del Friuli, Italy), and commercial low chromium ferritic stainless steel (SS 430),

supplied by Outokumpu (Espoo, Finland), were cut (2 cm × 1 cm × 0.1 cm), abraded with wet SiC paper to 4000 grit using ethanol as lubricant, and were then sonicated and dried in air prior to immersion tests. Table 1 presents the elemental composition of the stainless steel and carbon steel.

**Table 1.** Stainless steel 430 and carbon steel (B450C) compositions (wt.%), according to suppliers Outokumpu (Finland) and Pittini Group (Italy).

| Element (wt.%) | C | Cr | N | Cu | P | S | Fe | PRE [1] |
|---|---|---|---|---|---|---|---|---|
| SS 430 | 0.05 | 16.2 | - | - | - | - | Balance | 16 |
| B450C | 0.22 | - | 0.12 | 0.8 | 0.5 | 0.5 | Balance | - |

[1] PRE (Pitting Resistance Equivalent) value is a common tool in stainless steel design for predicting suscepti-bility to pitting corrosion [13,14,39], where higher values indicate higher resistance to pitting corrosion. PRE is calculated on the basis of the Cr, Mo, W, and N content of an alloy, and the most common form of PRE is PRE = %Cr + 3.3 (%Mo + 0.5%W) + 16%N. Other parameters are assumed to be constant, such as surface condi-tion, heat treatment history, possible precipitation of intermetallic phases, inclusion level, grain size, and variation of the surface metallurgy.

The model solution simulates non-carbonated concrete pores and was prepared from 1:1 wt./wt. mixture of Portland cement type I, produced by CEMEX (CEMEX, S.A.B. de C.V., San Pedro Garza García, N.L., México) and ultrapure deionized water (18.2 MΩ·cm). Table 2 gives the chemical composition of the cement and that of the cement extract solution after filtration, according to Wang et al. [23]. The mixture was stored for 24 h in a sealed container in order to allow cement hydration and avoid the absorption of $CO_2$ from the air. Subsequently, the suspended particles were removed from the supernatant by filtering the solution with a 125 mm pore-size filter paper (Whatman, Kent, UK). The experiments were carried out in the absence of chlorides, because the steel surface in concrete is not usually exposed to $Cl^-$ ions during the initial stages. The initial pH value of this model solution was 13, and it was regularly checked by a pH meter during the immersion of the steel samples.

**Table 2.** Composition of Portland cement according to the producer and chemical analysis of filtered cement extract solution as proposed [23] after 24 h.

| Compound | Portland Cement Weight % [2] | Cement Extract [3] Weight % [3] | Ion | mmol $L^{-1}$ |
|---|---|---|---|---|
| CaO | 66.84 (as $Ca^{2+}$) | 58.42 | $Ca^{2+}$ | 6.4 |
| $SiO_2$ | 21.35 | 22.30 | K+ | 35.1 |
| $Al_2O_3$ | 4.87 (as $Al^{3+}$) | 4.62 | $SO_4^{2-}$ | - |
| $Fe_2O_3$ | 2.89 (as $Fe^{3+}$) | 2.44 | $Na^+$ | 18.3 |
| $SO_3$ | 2.42 | 2.20 | $OH^-$ | 56.4 |
| MgO | 1.16 (as $Mg^{2+}$) | 1.92 | - | - |
| $K_2O$ | 0.39 | 0.35 | - | - |
| $Na_2O$ | 0.08 | 0.28 | - | - |

[2] According to this study. [3] According to [23].

### 2.2. Immersion Test and Surface Characterization

The triplicate steel samples (0.8 $cm^2$ working area) were immersed in 50 mL of ce-ment extract solution, according to the ASTM G31-12a standard for laboratory immersion corrosion testing of metals [40]. They were withdrawn after 720 h (30 days), rinsed with deionized water, and dried in air at room temperature (21 °C). The damage on the steel surface was evaluated on removal of the corrosion products formed after exposure for 30 days in cement extract solution. The carbon steel samples were submerged in a so-lution of hydrochloric acid and hexamethylene tetramine for 10 min at 21 °C, while SS 430 samples were submerged in a solution of nitric acid and hydrofluoric acid for 5 min at 21 °C, according to the cleaning procedure recommended by ASTM G1-03 standard [41].

Furthermore, their surfaces were characterized by Scanning Electron Microscopy (SEM-EDS, XL-30 ESEM-JEOL JSM-7600F, JEOL Ltd., Tokyo, Japan), and the corrosion products were analyzed with an X-ray Photoelectron Spectrometer (XPS, K-Alpha, Thermo Scientific, Waltham, MA, USA) equipped with a monochromatic Al K-alpha radiation source (1486.6 eV). The pass energy was 50 eV and the energy step size was 0.1 eV for the scan of XPS spectra, while for survey spectra, they were 100 eV and 1 eV, respectively. The XPS spectra were obtained after sputtering the specimens' surface with a scanning argon-ion gun during 15 s. The spectra were calibrated by setting the main line for the O 1s signal of oxygen in oxides at 530.2 eV, according to the procedures suggested by Yamashita and Hayes for transition metal oxides [42].

### 2.3. Electrochemical Measurements

An Interface-1000E potentiostat/galvanostat/ZRA (Gamry Instruments, Philadelphia, PA, USA) was used for electrochemical experiments, with a typical three-electrode cell configuration inside a Faraday cage. The working electrodes were steel plates (0.8 cm$^2$); the Pt plate was used as an auxiliary, and a saturated calomel electrode (SCE) was the reference electrode. All experiments were carried out at room temperature (21 °C). Electrochemical Impedance Spectroscopy (EIS) measurements were performed at the open circuit potential (OCP), applying an AC signal of ± 10 mV amplitude, in a frequency range from 100 kHz to 10 mHz, and with a sampling size of 10 data points/decade. EIS diagrams were recorded at different immersion periods: 10 min (initial) and 24, 168, 360, 504, and 720 h (30 days).

## 3. Results and Discussion

### 3.1. Steel Surface Characterization

Figure 1 shows the SEM images of ferritic SS 430 (Figure 1a) and carbon steel B450C (Figure 1b) surfaces of control samples, observed by low-angle backscattered electrons (LABE). The EDS elemental analysis (Table 3) confirms the presence of a high Cr content (≈16 wt.%) on the ferritic SS 430 surface, which plays an important role in increasing pitting corrosion resistance [7,15], as well as the presence of C (1.94 wt.%), N (1.14 wt.%), Mn (0.65 wt.%), and V (0.28 wt.%). Generally, Mn is added to stainless steels during melting to assist in de-oxidation and to prevent the formation of S-inclusions, which can cause hot cracking problems. Manganese allows hardness and strength to increase when it forms phases such as Mn3Cr and nitrides with Nitrogen (N). Vanadium provides a ferrite stabilizing effect, and it is used to help control the grain size of the steel, keeping it small, as stable compounds are formed in steel with Carbon ($V_6C_5$, carbide phase) and Nitrogen (VN, nitride phase), which block and prevent the grains from growing larger. Thus, this finer grain structure increases the ductility, hardness, and strength of steel. The existence of carbide phase $(Cr,Fe)_7C_3$ and Cr-nitride have also been reported [15,43].

**Table 3.** EDS surface analysis (wt.%) of SS 430 (Figure 2a) and carbon steel B450C (Figure 2b) control samples.

| Element | | C | Cr | Mn | Si | O | V | Cu | N | S | Fe |
|---|---|---|---|---|---|---|---|---|---|---|---|
| SS 430 | General | 1.83 | 16.32 | 0.65 | 0.6 | 0.34 | 0.28 | - | - | - | 79.97 |
| | A | **3.04** | **24.03** | - | 0.25 | 0.62 | 0.7 | - | **3.27** | - | 68.09 |
| | B | **17.19** | **9.21** | - | **28.86** | 1.35 | - | - | - | - | 43.39 |
| B450C | General | 2.36 | - | 0.81 | - | 0.47 | - | - | - | - | 96.36 |
| | C | **5.02** | - | **1.31** | 0.41 | 1.44 | - | 0.83 | - | **0.38** | 90.36 |
| | D | **8.49** | - | 0.73 | **6.46** | 0.46 | - | - | - | - | 83.85 |

Note: Even though the surface of both alloys were continuously sonicated after the polishing with SC paper, the higher content of Si (SiC) on the SS 430 surface is not well understood. Since Si has strong affinity for oxygen, after the polishing process, the outer surface layer was enriched in this element.

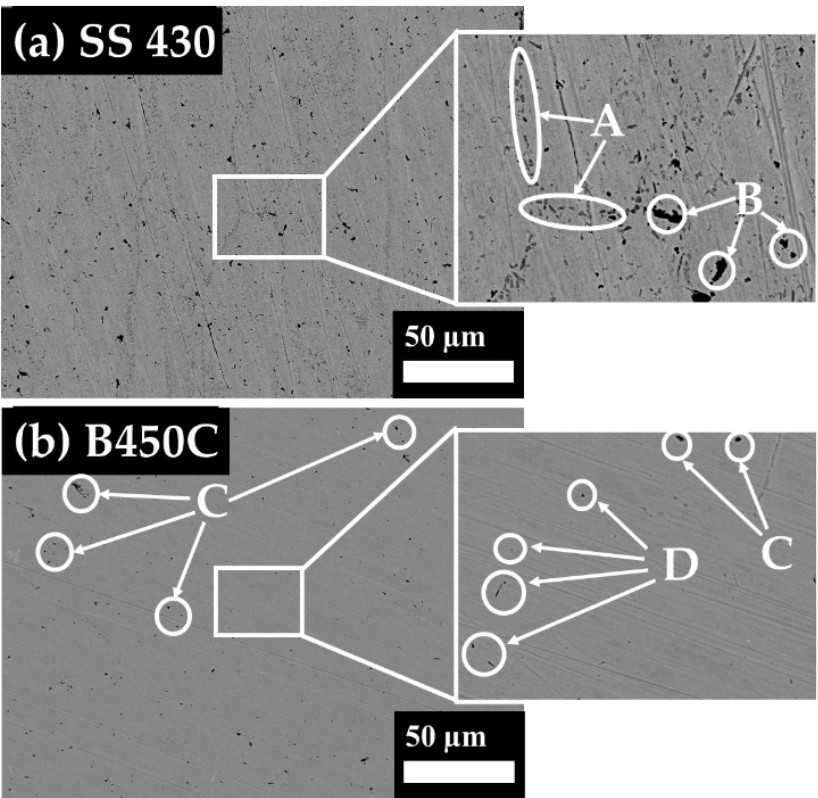

**Figure 1.** SEM images (500×) of surface steel control samples and magnification (3000×) of areas of interest: (**a**) SS 430 and (**b**) carbon steel B450C.

The SS 430 surface (Figure 1a) presents multiple gray dots (A), whose composition (Table 3) indicates that it is probably of chromium nitride and carbide phases, which typically precipitate at the grain boundaries because of the lower solubility of C and N, as well as being due to the fast diffusivity of Cr in the ferrite phase [43]. A small signal of V is also observed if this metal replaces the Cr sites in the lattice of Cr-C-N crystal structure, forming precipitates of vanadium carbonitrides (V (C, N)) [44]. There is also a darker area (B) that may be attributed to silicon carbide (SiC).

Carbon steel B450C (Figure 1b) contains as alloying elements mainly carbon (2.36 wt.%) and manganese (0.81 wt.%), according to the general chemical analysis EDS (Table 3). Black dots (C) with high Mn and lower sulfur (S) contents were also observed; this could be considered as the MnS phase, which has been reported for this type of steel [45,46]. However, due to the high carbon content (2.36 wt.), the phase of $Mn_3C$ is probably also present. The existence of Cu content may be attributed to the quality of the scrap used to produce this carbon steel [46]. Another zone (D) was also analyzed, which appears to be the SiC phase. Manganese and silicon, although not explicitly reported by the supplier, are always present.

In order to correlate the elemental quantification analysis SEM-EDS (Table 3) with the phases present on the studied steel surfaces, XPS of SS 430 and carbon steel B450C control samples was performed. Figure 2 shows the high-resolution spectra for Fe, O, Cr, and Mn. Based on the average of the binding energies [47], the spectra of Fe2p, Cr2p, and O1s signals were deconvoluted into chemical states, as the most probable oxidized and non-oxidized components for corresponding chemical assignments.

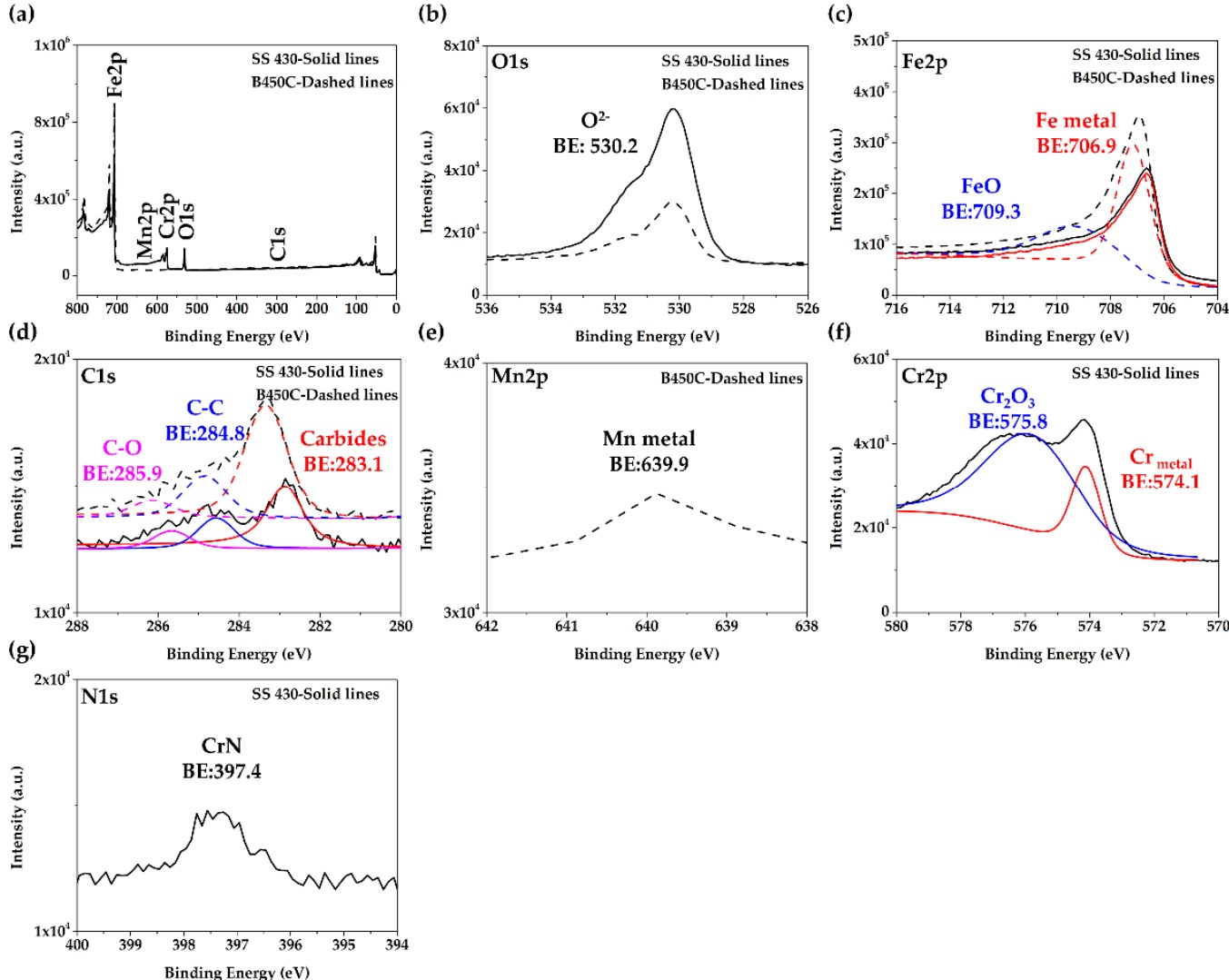

**Figure 2.** Overview of X-ray photoelectron spectroscopy (XPS) spectra acquired from SS 430 and carbon steel B450C control sample surfaces: (**a**) full spectrum; spectrum for (**b**) O1s, (**c**) Fe2p, (**d**) C1s, (**e**) Mn2p, (**f**) Cr2p, and (**g**) N1s.

The peaks of Fe (Figure 2c) and Cr (Figure 2f) were separated into different oxidation states [48]. The displayed peak for O1s corresponding to $O^{2-}$ was associated with an oxide phase, attributed to FeO (at 709.3 eV) and $Cr_2O_3$ (at 575.8 eV), formed on the surfaces in contact with the atmosphere, before the immersion in cement extract model solution. Both steel surfaces also present the Fe2p peak (at 706.9 ± 0.2 eV), corresponding to Fe-metal matrix. Furthermore, SS 430 shows the peak of Cr2p (at 574.1 eV), attributed to Cr as an alloying element (Figure 2f). Signals with lower intensity of carbon (Figure 2d) and nitrogen (Figure 2g) were also observed, corresponding to C-O (at 285.9 ± 0.2 eV) and C-C (at 284.8 ± 0.2 eV) bonds, usually reported as contamination, as well as the signals at 283.1 ± 0.2 eV and 397.4 eV, attributed to carbides and chromium nitrides, respectively [44]. The control sample of carbon steel B450C also displayed a very low signal of metallic Mn (Figure 2e, at 639.9 eV); however, the signals of S and MnS suggested by EDS were not detected.

### 3.2. Steel Surface Characterization after Exposure to Cement Extract Solution

Figure 3 shows SEM images of the steel surfaces studied after their exposure to the cement extract solution for 168 (7 days) and 720 h (30 days). The SS 430 surface (Figure 3a) exhibits two different zones (A and B) after 168 h of immersion. EDS analysis (Table 4)

suggests that zone A is associated with the metal matrix, having a slight increase in oxygen content (1.69 wt.%) with respect to the unexposed surface (Figure 1a and Table 3), and with the presence of traces of Ca. On the other hand, crystals (zone B) with high contents of C (13.75 wt.%), O (53.8 wt.%), and Ca (30.22 wt.%) were deposited on the stainless steel surface, and this analysis suggests the presence of $CaCO_3$ crystals. According to study [49], in neutral and alkaline pH, the $\zeta$-potential of stainless steel surfaces has negative values (between $-49.8 \pm 0.6$ mV and $-59.0 \pm 4.2$ mV), and as a consequence positive cations may be attracted, such as $Ca^{2+}$, present in the cement extract solution (Table 2). Results of employed atomistic simulation indicated that the most stable (001) surfaces of $CaCO_3$ are those of two hydrated phases, monohydrocalcite ($CaCO_3—H_2O$) and Ikaite ($CaCO_3—6H_2O$), which may exist an alkaline water environment [50]. However, after the longer period of exposure of 30 days (Figure 3c), it seems that most of the $CaCO_3$ crystals (zone B) detached from the SS 430 surface because of their mass. After the shorter period of exposure (168 h), the carbon steel surface of B450C (Figure 3b) showed B zones with crystals, having high content of O and Ca, attributed to $CaCO_3$ (Table 4). The C zones seem to represent the matrix of the steel, while the D zones of high O content may attributed to the iron-hydroxides layer, as a corrosion product. An evident change in the B450C morphology is notable in the last period of exposure of 30 days (Figure 3d), when most of the surface is covered by voluminous areas presenting high oxygen and iron content (zones D and E), as corrosion products. Only small areas (zone C) appear to be a part of iron steel matrix.

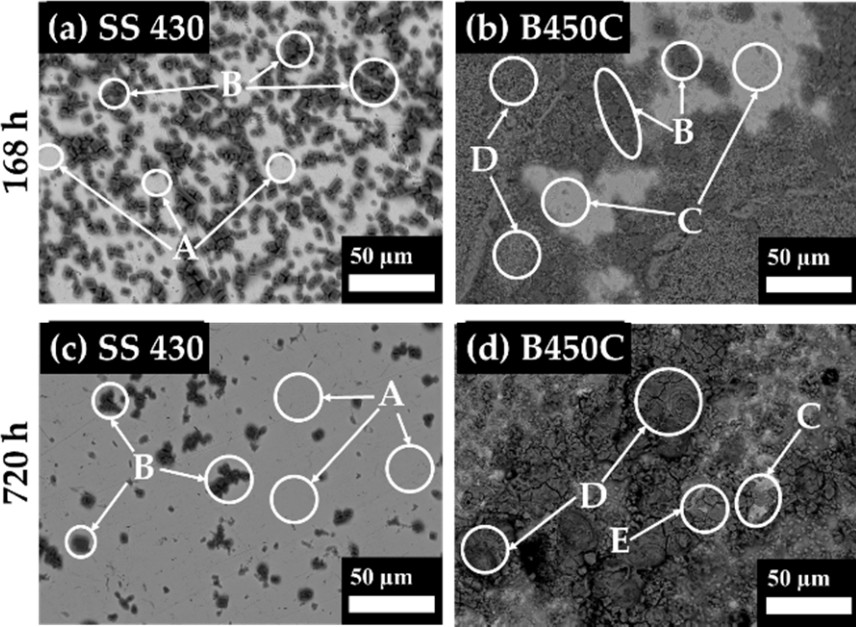

**Figure 3.** SEM images (500×) of steel surfaces after exposure to cement extract solution for 168 h: (**a**) SS 430 and (**b**) carbon steel (B450C); and for 720 h: (**c**) SS 430 and (**d**) carbon steel (B450C).

Figure 4 presents the XPS spectra of SS 430 surface exposed for 168 and 720 h. The O1s spectrum (Figure 4b) for both time periods shows contributions from two oxidation states, corresponding to $O^{2-}$ in oxides (at 530.2 eV) and $OH^-$ in hydroxides (at $531.5 \pm 0.2$ eV), respectively, attributed mainly to FeO (at $709.3 \pm 0.2$ eV, Figure 4c), $Cr_2O_3$ (at $576.1 \pm 0.1$ eV), and $Cr(OH)_3$ (at $577.7 \pm 0.1$ eV) (Figure 4e). The passive film of SS 430 is mainly composed of $Cr_2O_3$ and $Cr(OH)_3$. Carbon bonds (Figure 4d) may present the $CO_3^{2-}$ ion, as a part of $CaCO_3$ when the signal of Ca2p is registered (Figure 4f).

**Table 4.** EDS surface analysis (wt.%) of SS 430 and carbon steel B450C after exposure to cement extract solution for 168 h (7 days) and 720 h (30 days).

| Element | | | C | Cr | Mn | Si | O | Ca | Cu | Fe |
|---|---|---|---|---|---|---|---|---|---|---|
| SS 430 | 168 h | A | 2.87 | **16.84** | - | - | **1.69** | 0.8 | - | **77.8** |
| | | B | **13.75** | - | - | - | **53.8** | **30.22** | - | 2.49 |
| | 720 h | A | 1.81 | **16.43** | - | 0.44 | **1.06** | - | - | **80.01** |
| | | B | **13.75** | 4.62 | - | 0.37 | **32.87** | **21.3** | - | 20.92 |
| B450C | 168 h | B | **9.1** | - | 0.73 | - | **47.88** | **27.76** | - | 15.27 |
| | | C | 6.43 | - | 0.67 | - | **7.57** | 0.85 | - | **84.48** |
| | | D | 4.54 | - | 1.43 | 0.27 | **47.51** | 3.73 | - | **42.51** |
| | 720 h | C | 4.06 | - | 0.73 | 0.26 | **8.65** | - | 0.97 | **85.33** |
| | | D | 3.29 | - | 0.36 | - | **44.1** | - | - | **52.26** |
| | | E | 3.09 | 0.24 | 0.27 | 0.41 | **41.98** | - | 0.84 | **53.15** |

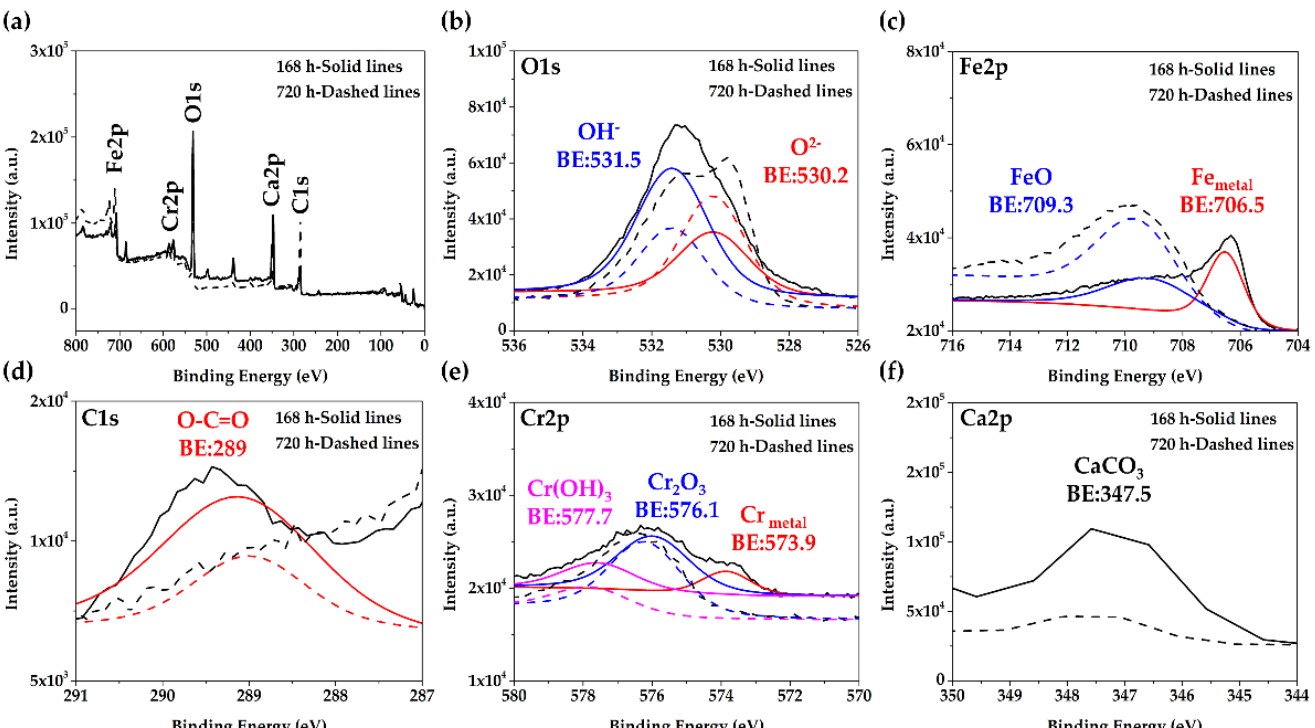

**Figure 4.** Overview of X-ray photoelectron spectroscopy (XPS) spectra acquired from SS 430 surface after 168 h and 720 h of exposure to cement extract solution: (**a**) full spectrum; spectrum for (**b**) O1s, (**c**) Fe2p, (**d**) C1s, (**e**) Cr2p, and (**f**) Ca2p.

The XPS spectra of B450C carbon steel exposed for 168 and 720 h in cement extract solution (Figure 5) similarly present the oxygen binding energies corresponding to $O^{2-}$ in oxides (at 530.2 eV) and $OH^-$ in hydroxides (at 531.5 ± 0.2 eV), respectively (Figure 5b). They are attributed mainly to FeO (at 709.7 ± 0.1 eV) and FeOOH (at 711.7 ± 0.1 eV), whose signals are observed in Figure 5c, as well as to MnO (at 641 eV), representing the signal of Mn2p (Figure 5e). Carbon bonds (Figure 5d) may indicate the $CO_3^{2-}$ ion, as a part of $CaCO_3$ when the signal of Ca2p is registered (Figure 5f).

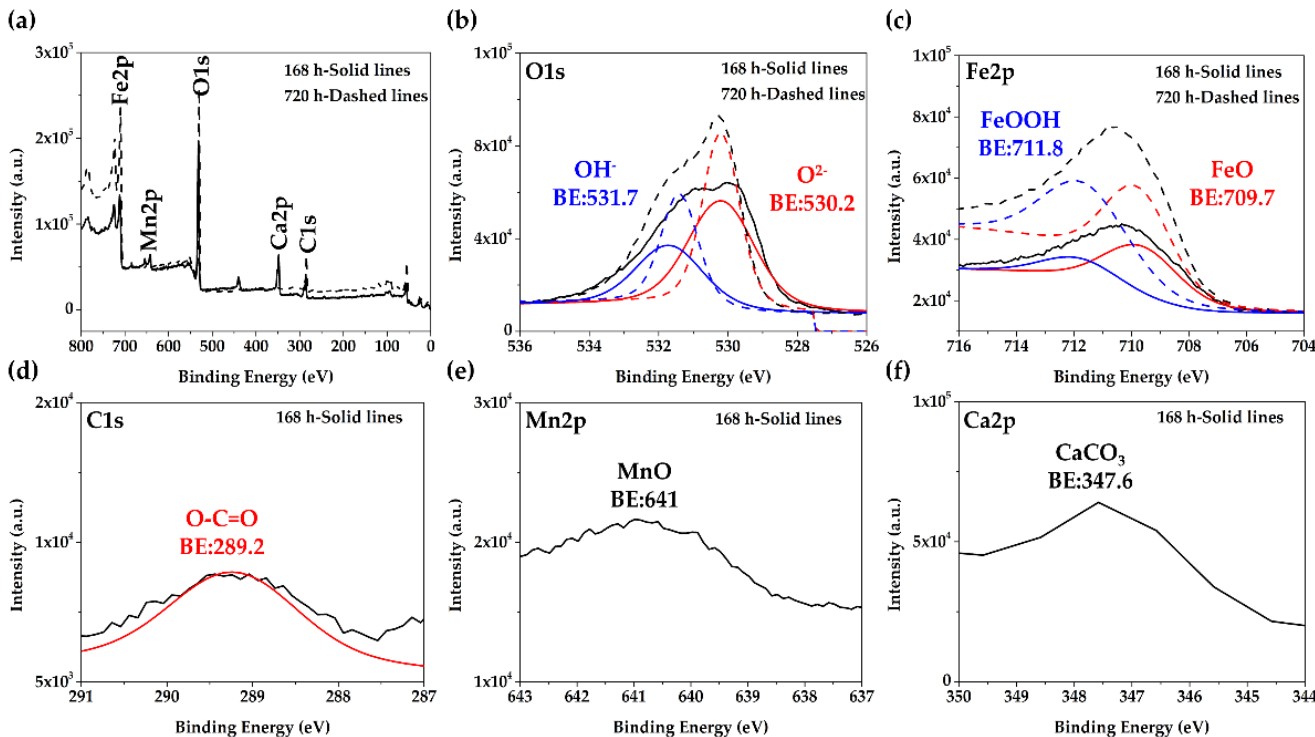

**Figure 5.** Overview of X-ray photoelectron spectroscopy (XPS) spectra acquired from carbon steel (B450C) surface after 168 h and 720 h of exposure to cement extract solution: (**a**) full spectrum; spectrum for (**b**) O1s, (**c**) Fe2p, (**d**) C1s, (**e**) Mn2p, and (**f**) Ca2p.

It may be concluded that the XPS registered spectra correlate well with the EDS-SEM analysis corresponding to both steels.

After the removal of the corrosion layers (Figure 6), two different particles in elemental composition were observed on the studied alloys. On the SS 430 steel surface, the particles (F in Figure 6a and Table 5), having high content of Cr, N, C, and V and which were also present initially on this surface (Figure 1 and Table 3), suggested Cr-C-N crystal structure, forming precipitates of vanadium carbonitrides V (C, N) [44]. In the meantime, on the carbon steel B450C surface, particles (G) with high Mn, Cu, and S contents, are present, as reported by the supplier and observed previously (Figure 1b and Table 3).

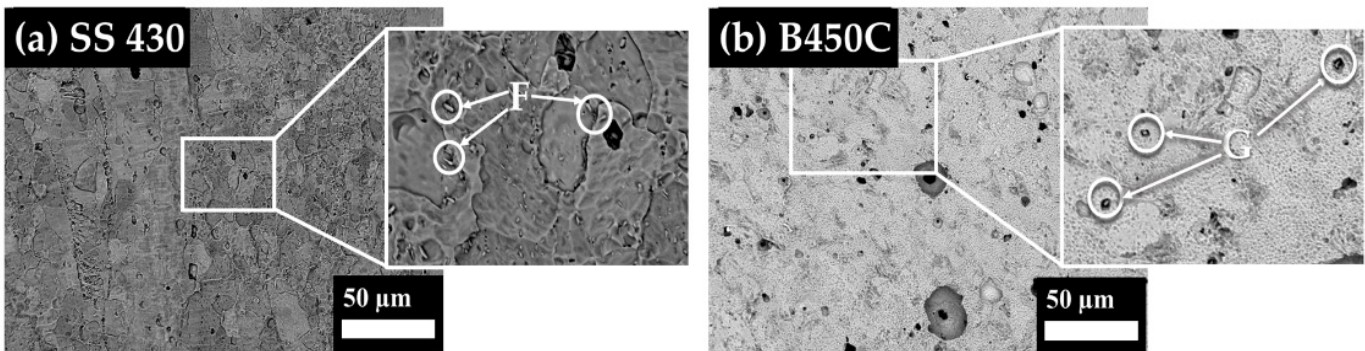

**Figure 6.** SEM images (500×) of steel surfaces after removal of corrosion layers formed during exposure to cement extract solution for 720 h: (**a**) SS 430 and magnification (3000×); (**b**) B450C and magnification (1500×).

**Table 5.** EDS surface analysis (wt.%) of SS 430 and carbon steel B450C after removal of corrosion layers formed during exposure to cement extract solution for 720 h.

| Element | | C | Cr | Mn | O | V | Cu | N | S | Fe |
|---|---|---|---|---|---|---|---|---|---|---|
| SS 430 | F | 8.22 | 43.1 | - | - | 1.65 | - | 13.54 | - | 33.5 |
| B450C | G | 3.98 | - | 30.54 | 1 | - | 4.64 | - | 10.42 | 49.44 |

Reported study [51] suggests that, according to the position in the galvanic series (potential values), the iron/steel would present anodic activity, while the trace metals, such as V (as precipitated V (C,N) carbonitrides), would be cathodic active areas. The following reactions are proposed, in addition to the cathodic reaction of $O_2$ reduction (Equation (1)):

$$Fe \rightarrow Fe^{2+} + 2e^-, \text{anodic reaction} \tag{2}$$

$$V^+ + e^- \rightarrow V, \text{cathodic reaction} \tag{3}$$

It is also considered that the inclusion of MnS does not dissolve when the localized corrosion (pitting) initiates; however, the dissolution of the iron matrix occurs from the immediate surroundings of these inclusions, which may give rise to the initiation and development of pits [52], reaching large size and depth. At the later stages, in the presence of local acidification ($H^+$), MnS may begin to dissolve (Equation (4)). The concentration increase of $H_2S$ in turn may contribute locally to additional cathodic processes (Equation (5)), and both reactions (Equations (3) and (5)) will accelerate the anodic dissolution (corrosion) of Fe (Equation (2)) [52].

$$MnS + 2H^+ \rightarrow H_2S + Mn^{2+}, \text{anodic reaction} \tag{4}$$

$$2H_2S + 2e^- \rightarrow 2HS^- + 2H_2, \text{cathodic reaction} \tag{5}$$

*3.3. Change in Time of pH of Cement Extract Solution during Exposure of Steel Samples*

Figure 7 presents the change in time of pH of the cement extract solutions during the exposure of SS 430 and carbon steel B450C samples for 720 h (30 days). The initial value of pH = 13 was kept approximately constant up to 168 h (7 days), decreasing abruptly to pH~9, and maintaining almost this value until the end of the experiment (720 h). The decrease in pH may be due primarily to the fact that carbon dioxide from the environment dissolved in the electrolyte, the volume of which is 50 mL in this experiment, promoting the formation of carbonic acid ($H_2CO_3$) [19]. It is considered that for 7 < pH < 10, the $HCO_3^-$ ions are the predominate species, which are corrosive to metals; thus iron releases ions ($Fe^{2+}$) that attract the $OH^-$ from the medium, mainly forming iron hydroxide II [53], which further lowers the pH.

*3.4. Change in Time of Corrosion Potential (OCP) of SS 430 and Carbon Steel (B450C) during Exposure in Cement Extract Solution*

Figure 8 presents the change in time of the average values of corrosion potential (open circuit, OCP, vs. SHE) of SS 430 and carbon steel (B450C) samples during their exposure for 720 h (30 days) in cement extract solution. The initial values are negative, between −192 mV (SS 439) and −150.2 mV (carbon steel B450C), showing a tendency to shift to less negative (positive) values and indicating the formation of passive layers on both steel surfaces. However, after 168 h, when pH of the electrolyte decreased abruptly to 9, the OCP of the carbon steel returned suddenly to a very negative value (−450 mV, above the initial one) until the end of the experiment (30 days), because the surface lost passivity state at a pH lower than 12. According to the Pourbaix diagram, the iron (Fe) is susceptible to corrosion when the pH value is below 11.5 [17]. On the other hand, the OCP of SS 430 continued reaching more positive values (Figure 8), even the pH~9, as an indication that the steel maintained its the passive state, contributed mainly to chromium (Cr) as alloying element.

Chromium migrates to the SS surface, reacting with the dissolved air oxygen in the water, and Cr-oxide is formed. A surface having enough Cr-reach sites develops very thin layers of a few atoms, creating the passive state on the SS surface [13,14,39].

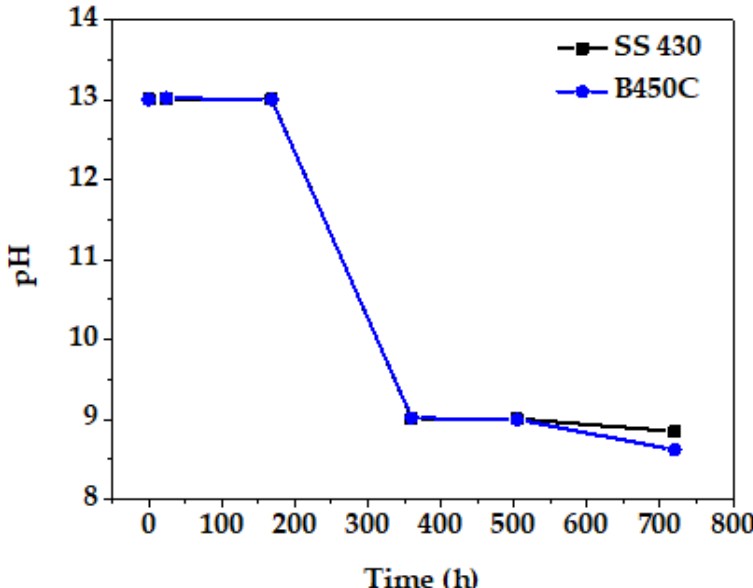

**Figure 7.** Change in time of pH cement extract solutions during the immersion of SS 430 and carbon steel B450C samples for 720 h (30 days).

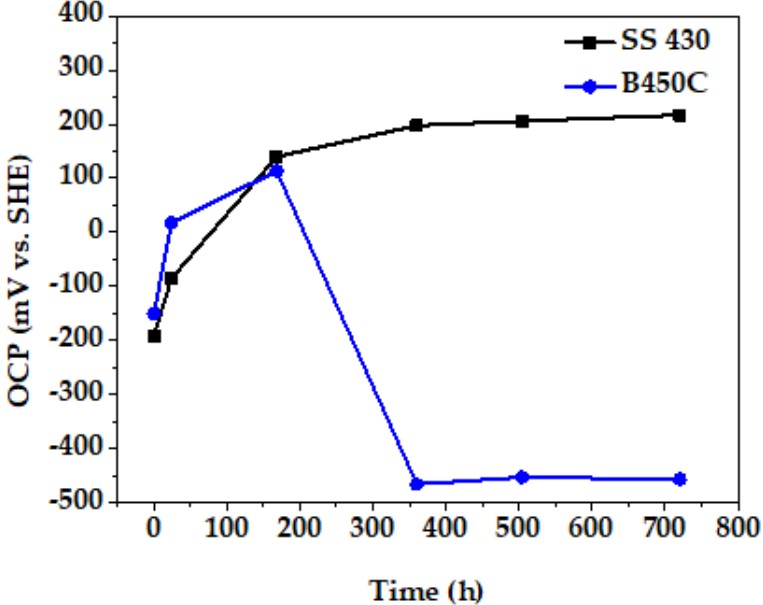

**Figure 8.** Change in time of open-circuit potential values (mV vs. SHE) of SS 430 and carbon steel B450C samples immersed in cement extract model solution (at 22 °C) up to 720 h (30 days).

*3.5. Electrochemical Measurements*

Electrochemical Impedance Spectroscopy (EIS)

Figures 9 and 10 present Nyquist and Bode EIS diagrams for SS 430 and carbon steel B450C samples exposed for different periods (up to 720 h) to cement extract solution. The Nyquist plots (Figure 9) show that, at initial time of exposure, each steel presents capacitive behavior, usually attributed to charge transfer and mass transport. However, after the first 24 h, the behavior changes to semi-linear diffusion impedance, caused by inductive

reactance of the metallic surface, and a linear slope with an angle of a $\approx 45°$ is observed. This is associated with diffusion control of the corrosion process, because of the passive films formed on steel surfaces.

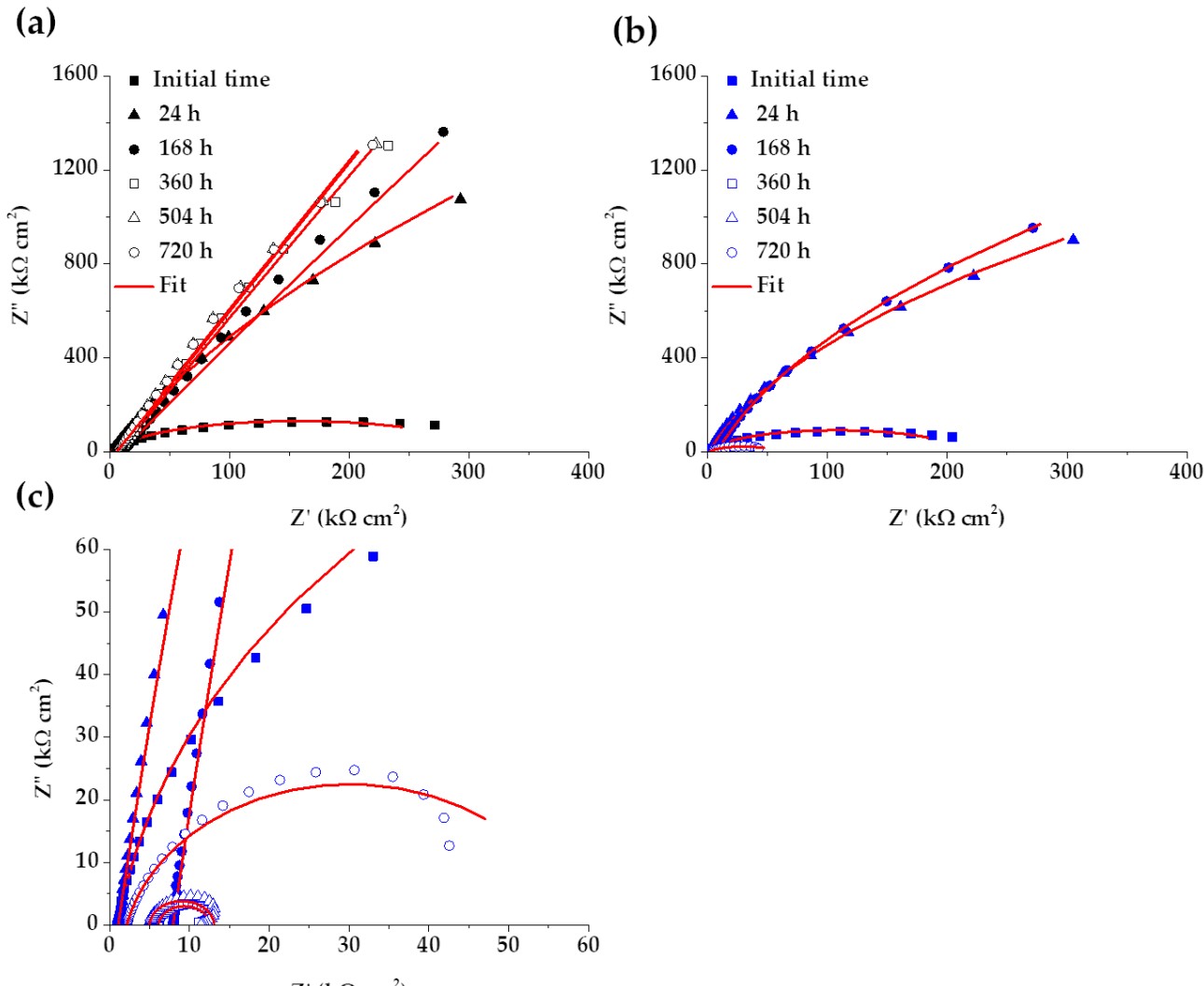

**Figure 9.** EIS Nyquist diagrams with respective fitting line for SS 430 and carbon steel B450C after different times of immersion in cement extract solution: (**a**) SS 430, (**b**) B450C, and (**c**) magnification of B450C (**b**) diagram.

The increase of the positive corrosion potential (OCP) values of SS 430 (Figure 8) indicate that the passive state is not influenced by the changes in pH of the cement extract solution (Figure 7). Thus, the immersion time of SS 430 improves its the corrosion resistance, maintaining firmly the diffusion impedance until the end of the experiment (Figure 9a), as typical for passivated stainless steel surfaces [48,54]. On the other hand, the carbon steel B450C keeps the diffusion impedance no longer than 168 h (Figure 9b) and when the passive state is lost a return to capacitive behavior is observed, because of the decrease of the pH (Figure 7). Consequently, the carbon steel displays semi-circles, whose diameter decreases with the time of immersion (Figure 9c).

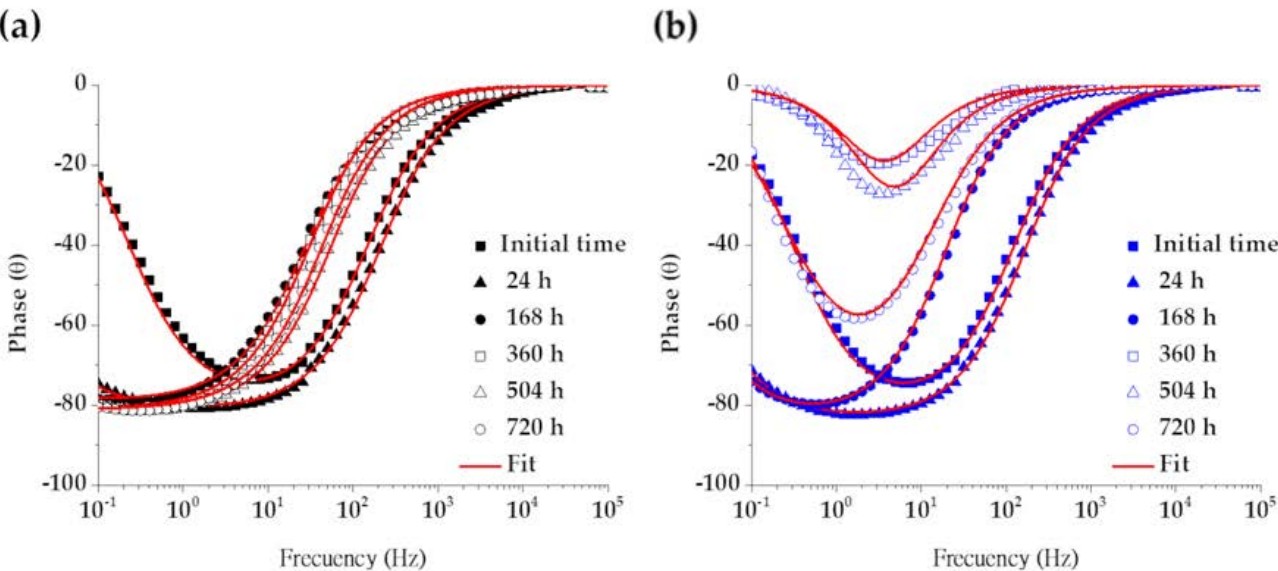

**Figure 10.** EIS Bode diagrams—Phase angle with respective fitting line for SS 430 and carbon steel B450C after different times of immersion in cement extract solution: (**a**) SS 430, (**b**) B450C.

The changes in the behavior of the Nyquist diagrams of the steels during immersion in cement extract solution are confirmed by the phase angle Bode diagrams (Figure 10). It may be seen that the SS 430 keeps an angle $\approx -80°$ until the end of the experiment (Figure 10a), while the carbon steel (Figure 10b) showed a tendency toward very low angles, which generally characterize capacitive surfaces [54]. A phase angle of $-90°$ indicates an electrode interface that is capable of accumulating electrical charges, avoiding migration of aggressive species like $O^{2-}$ and $Cl^-$ from the solution through interface, as also providing a high resistance to the corrosion process development, usually because of an existing stable and almost perfect passive surface layer [54–56].

Equivalent circuits (ECs) were proposed (Figure 11) to investigate the EIS data further. The first EC consists of a simplified Randles circuit (Figure 11a), where $R_s$ corresponds to the resistance of the electrolyte, $CPE_2$ is the capacitance of the double layer of the electrolyte/electrode interface, and $R_{ct}$ is the charge transfer resistance of the film. This type of model describes the electrochemical reactions of a passive system with only one time constant, which is consistent with other work regarding carbon steel in alkaline solutions [5,28]. The second proposed circuit (Figure 11b) has been used to fit different types of stainless steel in alkaline solutions [57–59]. The circuit is composed of a resistance at high frequencies, associated with the resistance of the solution ($R_s$). The resistance ($R_{cp}$) and capacitance $CPE_1$ are associated with the layers of passive $Cr_2O_3/Cr(OH)_3/FeO$ films and their hydroxide corrosion products. On the other hand, the resistance ($R_{ct}$) is associated with charge transfer resistance of the corrosion process and a capacitance $CPE_2$ related to the double capacitive layer in the corrosion-localized area. To obtain a better fit, capacitors may be replaced by constant phase elements (CPEs), which have an exponential factor, n, in the range from 0 to 1, where $n = 1$ for an ideal capacitor and $n = 0$ for an ideal resistor [57].

It can be noted (Table 6) that the $R_{sol}$ value increases over time, and one can speculate that this fact is closely related to the diminishing in time of pH of the cement extract solution (Figure 7). Therefore, a change of ion composition occurs at the interface metal/solution, because of more significant release of iron ions ($Fe^{2+}$), as also appearance of $HCO_3^-$ ions after the dissolution of $CO_2$ from air and an increase of $H_2S$ concentration (Equation (4)).

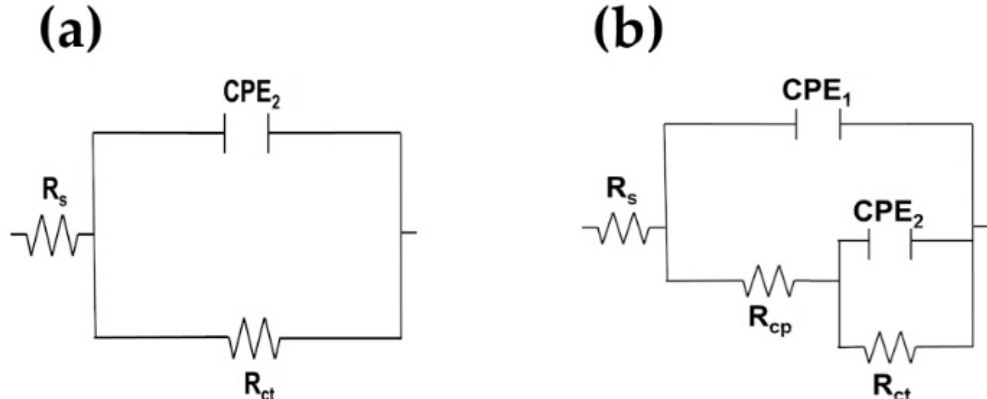

**Figure 11.** Equivalent circuits proposed for (**a**) carbon steel B450C and (**b**) SS 430.

**Table 6.** Fitting parameters obtained from the EIS measurements.

| | Time h | $R_{sol}$ kΩ cm² | $R_{cp}$ kΩ cm² | $CPE_1$ μS sⁿ cm⁻² | $n_1$ | $R_{ct}$ kΩ cm² | $CPE_2$ μS sⁿ cm⁻² | $n_2$ | $R_p$ kΩ cm² | $c^2$ $10^{-4}$ |
|---|---|---|---|---|---|---|---|---|---|---|
| SS 430 | 0.5 | 0.99 | 0.04 | 74.24 | 0.88 | 312.88 | 27.99 | 0.89 | 312.91 | 6.11 |
| | 24 | 1.15 | 1.99 | 285.75 | 1.00 | 10,766 | 16.54 | 0.90 | 10,768 | 3.23 |
| | 168 | 9.39 | 2.50 | 111.65 | 1.00 | 172,630 | 13.23 | 0.88 | 172,630 | 10.1 |
| | 360 | 7.65 | 2238.8 | 0.20 | 1.00 | 375,630 | 14.41 | 0.90 | 377,860 | 4.42 |
| | 504 | 4.45 | 0.94 | 314.38 | 1.00 | 1,570,000 | 14.58 | 0.90 | 1,570,000 | 6.63 |
| | 720 | 5.54 | 4,741,300 | 0.04 | 1.00 | 8,997.5 | 14.59 | 0.90 | 4,750,200 | 7.90 |
| B450C | 0.5 | 0.95 | - | - | - | 213.50 | 32.05 | 0.91 | 213.50 | 8.24 |
| | 24 | 1.05 | - | - | - | 4766.3 | 20.39 | 0.93 | 4766.3 | 3.00 |
| | 168 | 7.91 | - | - | - | 6331.3 | 19.65 | 0.93 | 6331.3 | 2.55 |
| | 360 | 6.08 | - | - | - | 6.53 | 156.63 | 0.95 | 6.53 | 12.6 |
| | 504 | 4.96 | - | - | - | 8.15 | 107.95 | 0.96 | 8.15 | 22.0 |
| | 720 | 2.12 | - | - | - | 55.90 | 128.00 | 0.86 | 55.90 | 8.62 |

The values of the fitting parameters obtained from the EIS measurements are presented in Table 6 and their fit $c^2$ ($10^4$) was good in the most cases.

The polarization resistance ($R_p$) values over time (Table 6) were calculated using the following equation [48]:

$$R_p = R_{cp} + R_{ct} \tag{6}$$

The values of $R_p$ for SS 430 increase over time, as an indication that the passive layer on its surface seems to be highly protective, mainly due to $Cr_2O_3$. For carbon steel, $R_p$ reaches its maximum value after 168 h and then decreases to minimum values, slightly changing until the end of the experiment.

Figure 12 presents the evolution of the passive layer thicknesses ($d$) with the immersion time. For the $d$ calculation, the $CPE_2$ values were used, transformed into the corresponding capacitance values according to the Brug formula (Equation (7)) [54]. The thickness was calculated from Equation (8) [60], where $\varepsilon_0$ is the vacuum permittivity ($8.85 \times 10^{-14}$ F cm⁻¹) and $\varepsilon$ is the dielectric constant of the passive film, which can be assumed as 15.6 for stainless steels [61,62].

$$C = CPE^{\frac{1}{n}} \left( \frac{R_s R_{ct}}{R_s + R_{ct}} \right)^{\frac{1-n}{n}} \tag{7}$$

$$d = \frac{\varepsilon \varepsilon_0 A}{C} \tag{8}$$

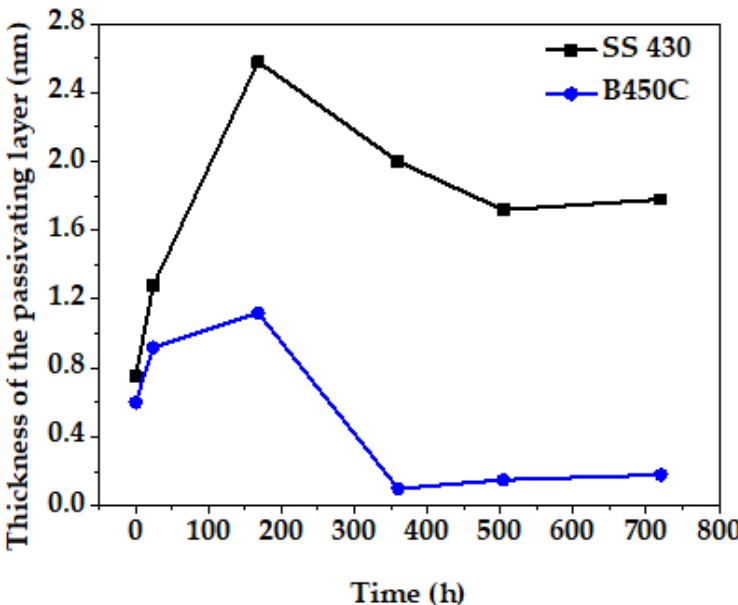

**Figure 12.** Passive layer thickness (*d*) evolution in the immersion time for SS 430 (black) and carbon steel B450C (blue).

The calculated values of thickness (*d*) revealed that they depend on the exposure time and the steel nature (Figure 12). For stainless steel 430, the thickness of the passive layer stabilizes at approximately 2 nm and remains at about this value until the end of the test (30 days). The passive film thickness values are similar to other values reported for the passive film formed on stainless steels exposed to low chloride content in solutions (0.5–6 nm); austenitic stainless steel (sanicro28) in 50 wt.% $H_3PO_4$; SS 316L in 0.1 M and 0.6 M NaCl; Fe15Cr alloy in 0.5 M $H_2SO_4$; and Fe10Cr and Fe20Cr alloys in 1 M NaOH [63–66]. In contrast, the passive layer of carbon steel B450C (Figure 12) tends to disappear after 360 h, which coincides with the decrease in the pH value of the cement extract solution (Figure 7), when the steel lost the passive state. However, when the surface was still passivated (up to 168 h), the greatest thickness was approximately 1.1 nm, similar to that estimated from complex capacitance plots (1.3–2.5 nm) for C15 mild steel in 0.1M NaOH alkaline solution [26].

## 4. Conclusions

The corrosion activities of commercial low chromium SS 430 ferritic and carbon steel grade B450C were studied during their exposure to cement extract model solution for 720 h (30 days). Before the immersion, SEM-EDS analysis suggested the presence of Cr (C-N) and V(C,N) phases as well as SiC on the SS 430 surface and a low content of Mn, while on the B450C surface the phases of SiC, MnS, and Mn3C were probably present. XPS analysis revealed that after the exposure, Fe-oxyhydroxide and Cr-hydroxide were formed as corrosion products, as well as $CaCO_3$.

The initial cement extract solution value of pH = 13 decreased abruptly to pH~9 after 168 h (7 days) of immersion of each steel, affected by $CO_2$ dissolution from air (in 50 mL model solution), maintaining close to this value until the end of the experiment (720 h). Consequently, the open circuit values (OCP) of the carbon steel shifted to very negative (loss of the passive state), while the SS 430 OCP values were positive (no passive state changes).

The Nyquist plots of EIS showed that, at the initial time of exposure, each steel presented capacitive behavior, attributed to charge transfer and mass transport. After the first 24 h, the behavior changed to semi-linear diffusion impedance, associated with diffusion control of the corrosion process, because of the formed passive films on steel surfaces. However, the SS 430 maintained this behavior steadily until the end of the

experiment (30 days), while the carbon steel B450C returned to capacitive behavior after 168 h, when the passive state was lost because of pH change of the solution. The phase angle Bode diagrams confirmed these changes, keeping the SS 430 angle of $\approx -80°$ until the end, while the carbon steel showed a tendency to very low angles, reaching $-1°$, generally characteristic of capacitive surfaces

Two equivalent circuits (ECs) were proposed to investigate further the EIS data and describe the electrochemical activity of the studied steels.

The calculated values of $R_p$ for SS 430 increased over time as an indication of a highly protective passive layer, while for carbon steel $R_p$ reached its maximum value after 168 h and then decreased, maintaining minimum values approximately five orders lower than those of the stainless steel.

The calculated thickness ($d$) of the passive layers revealed that they depend on the exposure time in cement extract solution and the nature of the steel. For SS 430, the thickness of the passive layer stabilized at about 2 nm and remained constant until the end of the test (30 days). In contrast, the passive layer on carbon steel B450C disappeared after 360 h ($d \approx 0.1$ nm) because of lost state of passivity.

**Author Contributions:** Á.B. performed the preparation of samples and the corrosion tests. Á.B. and L.V. discussed the results and wrote the manuscript. S.F.J. contributed for EIS data analysis. M.C. and S.L. supervised the project. All correspondence should be addressed to L.V. All authors have read and agreed to the published version of the manuscript.

**Funding:** This research received no external funding.

**Data Availability Statement:** Data present in this study are available on request from the corresponding author. The data are not publicly to privacy issues.

**Acknowledgments:** Ángel Adrián Bacelis Jiménez acknowledges the Mexican National Council for Science and Technology (CONACYT) for scholarship granted to him for his Ph.D. study and for research stay at Department of Engineering and Applied Sciences of Bergamo University of Italy. The authors gratefully thank to the National Laboratory of Nano- and Biomaterials (LANNBIO-CINVESTAV) for allowing the use of SEM-EDS and XPS facilities, also to Victor Rejón Moo and Wilian Cauich for their support in data acquisition.

**Conflicts of Interest:** The authors declare no conflict of interest.

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
