# Peer review of "Corrosion Activity of Carbon Steel B450C and Low Chromium Ferritic Stainless Steel 430 in Cement Extract Solution"

_buildings, doi:10.3390/buildings11060220_

Round 1
Reviewer 1 Report
The purpose of this article was to investigate the corrosion behaviour of two types of steel (carbon B450C and SS 430) immersed in cement extract solution.
It is well known that stainless steel will have much better corrosion protection than carbon steel. Therefore, I wondered about the advisability of undertaking this research topic. However, the choice of measurement techniques, broad spectrum of corrosion testing and the consistent results of these analyses are interesting. So, the article may be considered for publication in the “Bulidings” journal.
The structure of the article, conducted research work and selected references, were carried out properly.
In principle, in my opinion, a few changes are necessary to be done and following comments/suggestion should be taken into consideration:
157 line – it will be worth to add, that Fig.1 present images of surfaces before immersion in cement or as a control samples.
177 and 185 line - whether the high SiC content on the surface of SS 430 is not related to impurities created during the preparation of the specimens during metallographic preparation (grinding and polishing with SiC paper)?
187 line – in will be better (more visible) if the magnification of Fig 1a and Fig 1b will be the same size as 500x images, and Fig 1b will be under Fig 1a.
324 line – I believe, that it would be better if the results of Table 6 were presented in the form of a graph, the fluctuation of OCP values for steel B450C would be clearly visible.
In addition, I found some minor errors in the text:
273 line – worf “of” is not necessary.
395 line – at circa instead “at cerca”
432 line – chromium instead crhomium
440 line – “affected by air CO2 dissolution” – is not clear. It should be rather: affected by CO2 dissolution from air?

Author Response
Reviewer 1 reports: the introduction provide sufficient background and include all relevant references; (2) the research is designed appropriately; the methods are described adequately and many different types of tests were performed and some good results were obtained and discussed.
The structure of the article, conducted research work and selected references, were carried out properly. The choice of measurement techniques, broad spectrum of corrosion testing and the consistent results of these analyses are interesting. So, the article may be considered for publication in the “Bulidings” journal.
Responses: All comments/suggestion have been taken into consideration and we appreciate very much Reviewer´s comments.
Lines 162-163: Figure 1 shows the SEM images of ferritic SS 430 (Figure 1a) and carbon steel B450C (Figure 1b) surfaces of control samples, observed by ……
Lines 197-198: Note: Even the surface of both alloys were continuously sonicated after the polishing with SC paper, the higher content of Si on the SS 430 surface is not well understandable. Since Si has a strong affinity for oxygen, the outer layer was enriched of this element, after polishing process
Fig 1b is now located under Fig 1a.
Table 6 is presented now as Figure 8 (OCP values)
Lines 396-397: affected by CO2 dissolution from air.

Reviewer 2 Report
The manuscript is well written. The experiments have been well organized and conducted and the obtained results are supported by the proposed mechanisms.
The manuscript can be accepted only after minor revision for the following reasons:
The magnification as well as resolution of Figure 1 should be improved.
An explanation of the absence of the corresponding time constant for the claim "the impedance response is associated with the conductivity of the passive film (FeO/FeOOH), and growth mechanism of the passive film has little relation to time" should be provided.
The added value of the present work as well as the innovation should be highlighted.
Author Response
Reviewer 2 reports: The manuscript is well written. The experiments have been well organized and conducted and the obtained results are supported by the proposed mechanisms. The manuscript can be accepted only after minor revision for the following reasons
Responses to Rev.2 comments:
Figures 1 (a) and (b) present zooms (3000x) of areas of interest and images now are more visible.
Rev.2: An explanation of the absence of the corresponding time constant for the claim "the impedance response is associated with the conductivity of the passive film (FeO/FeOOH), and growth mechanism of the passive film has little relation to time" should be provided.
Response: Thank you for the comment. This statement comes from reference [59], where it is commented for a model with only one time constant: “In this model, the impedance response is associated with the conductivity of the passive film, and growth mechanism of the passive film has little relationship with the time. ”
[59] H. Luo, H. Su, C. Dong, X. Li, Passivation and electrochemical behavior of 316L stainless steel in chlorinated simulated concrete pore solution, Appl. Surf. Sci., 400 (2017), pp. 38-48, 10.1016/j.apsusc.2016.12.180
The paragraph in our text is confusing and really does not fit what we really want to comment about the model and it has been deleted, and replaced by:
Lines 377-379: “This type of model describes the electrochemical reactions of a passive system with only one time constant which is consistent with other work regarding carbon steel in alkaline solutions [5,28]”.
Rev.2:The added value of the present work as well as the innovation should be highlighted.
Response: we highlighted the value of our research, as a part of last paragraph of the introduction (as the goal of our study).
Lines 83-96: This research compares the corrosion activity of commercial Italian carbon steel to that of low chromium ferritic Finland stainless steel, exposed for 30 days in cement extract unbuffered solution, in order to simulate the concrete environment at the steel-concrete-pore interface. Both steels have been proposed as reinforcement in a lower pH of concrete, than that of the traditional pH of Portland cement-concrete, in the presence of binders. However, it is very important previously to be established the corrosion activity of each steel when pH in the traditional concrete-pore environment changes in time. Applying a variety of different techniques and methods help to contribute in this aspect. Two non-destructive electrochemical techniques were performed, viz: free corrosion potential monitoring at open circuit potential (OCP) and electrochemical impedance spectroscopy (EIS). The surfaces of the steels were characterized by scanning electron microscopy (SEM) and X-ray photoelectron spectroscopy (XPS) techniques. In our knowledge, no other research on this topic area has been previously undertaken.

Reviewer 3 Report
The paper is clearly structured and quite understandable. Thanks for sharing your findings. However, there is some clarification necessary:
Table 7. shows fitting results of EIS data, wich are very questionable. No information on reproducibility is made. Looking at shifting values for Rsol, the whole fitting cannot be really explained. Why does it increase until 168 h followed by a decrease? Pore solution should remain constant during experiment. Please re-write this part with more detailed explanations.
Chapter 3.5.2 talks about EC-Noise measurements for PI. Conclusions drawn are very questionable. This part shall be removed, since no proof is given that pitting corrosion occured at the specimens investigated. Within such artificial pore solution no pitting would be expected if no chlorides are involved.
In order to investigate corrosion stage linear polarization or galvanostatic pulse would provide better indicators.
Language needs some improvement, especially in terms of spelling and right use of tense.
Please keep numbers and units together using fixed space (shift+ctrl+space). At some places space is missing.
Author Response
Reviewer 3 reports: The introduction proved sufficient background and includes all relevant references. The paper is clearly structured and quite understandable. Thanks for sharing your findings. However, there is some clarification necessary.
Table 6 (new number) shows fitting results of EIS data, which are very questionable. No information on reproducibility is made. Looking at shifting values for Rsol, the whole fitting cannot be really explained. Why does it increase until 168 h followed by a decrease? Pore solution should remain constant during experiment. Please re-write this part with more detailed explanations.
Response: Thank you for the comment.
Authors reported that the surface chemistry of steel is governed by many factors, such as the ion composition of the pore solution and its change in time and the degree of pore saturation [B]. These changes resulted in a different passivation process and a thicker passive film with a higher proportion of Fe2+ oxides.
[B] H. Zheng, C.S. Poon, W. Li, Mechanistic study on initial passivation and surface chemistry of steel bars in nano-silica cement pastes, Cem. Concr. Compos., 112 (2020), Article 103661.
Thus, one can speculate that some change in time of pH of the cement extract solution (Figure 7) may contribute to change in the ion composition at the interface metal/electrolyte and therefore, Rsol value does not remain constant during the experiment. In the text, a new paragraph is included.
Lines 392-398 (after Figure 11):
“It can be noted (Table 6) that the Rsol value increases over time and one can speculate that this fact is closely related with the pH diminishing of the cement extract solution and change of ion composition at the interface metal/solution. It was mentioned before that the local acidification [H+) of the solution may involve the presence of new ions, such as: iron (Fe+2) released ions; HCO3- ions as predominate species (after the dissolution of CO2 from air) and the concentration increase of H2S (due to the dissolved MnS, Equation 4).”
Rev.3: Table 7 (new number Table 6) shows fitting results of EIS data, which are very questionable
Response: The adjusted values resumed in Table 6 have good fit c2 (104) in the most cases and it is observable in EIS diagrams.
Rev.3: Chapter 3.5.2 talks about EC-Noise measurements for PI. Conclusions drawn are very questionable. This part shall be removed, since no proof is given that pitting corrosion occured at the specimens investigated. Within such artificial pore solution no pitting would be expected if no chlorides are involved.
Response: The chapter 3.5.2 was removed. The goal of the presented results was to compare the corrosion attack on the alloy surfaces observed even in the absence of chloride. The fluctuation of corrosion current or potential (at OCP), considered as EN, is a common method for characterization of uniform or localized corrosion of metals, not necessary in the presence of pitting attacks.

Round 2
Reviewer 3 Report
Thanks for considering my comments.
For final submission, please check carefully line breaks within words and prevent them between numbers and units.
This manuscript is a resubmission of an earlier submission. The following is a list of the peer review reports and author responses from that submission.
Round 1
Reviewer 1 Report
Comments and Suggestions for Authors:
This manuscript presents a comparative study about the corrosion of carbon steel B450C and stainless steel 430 in cement extract solution. The results show that SS 430 presents less corrosion activity than carbon steel B450C. Many different types of tests were performed and some good results were obtained and discussed.
However, the following are the items that the authors would need to address:
- In line 22, the word “versality” seems like misspelled.
- The title of the manuscript is “Corrosion activity of …….”. My suggestion is to revise the title to make is more explicit and meaningful.
- In figure 2, there is only one the dashed line in (e) and only solid line in (g). But in these two figures, it is written “SS 430-Solid lines B450C-Dashed lines”. Where is the other one? The same thing happens to figure 5(d)(e)(f).
- In figure8, the rang of the abscissa and the ordinate should be the same. And based on this, the authors can analyze the “linear slope with an angle of a=45º”. So in this part, the authors are suggested to carefully analyze.
- In 366 line, the authors say that the Rcp and CPE1 are associated with the layers of corrosion products. In my viewpoint, there are more related to passive film for SS.
- The conclusions part is too long and contains too much details. The authors are suggested to carefully summarize the conclusions.
Reviewer 2 Report
Having studied the obtained material and without receiving additional explanations, we note that the article does not contain anything new - neither in test methods, nor in corrosion mechanisms or behavior features of these materials. Consequently, publication of the article does not seem necessary. Perhaps the purpose of the article was something else, but this is not indicated in the presented material. It should be noted 1) the results of scanning spectroscopy research are related to the topic of the article - the compositions of all non-metallic inclusions are described - information on the text is not used (Fig. 1, table 3) 2) the calculation of the thicknesses of passive films is not justified 3) cathodic reactions need to be checked